# Urinary Oxidative Damage Markers and Their Association with Obesity-Related Metabolic Risk Factors

**DOI:** 10.3390/antiox11050844

**Published:** 2022-04-26

**Authors:** Salah Gariballa, Abderrahim Nemmar, Ozaz Elzaki, Nur Elena Zaaba, Javed Yasin

**Affiliations:** Internal Medicine, Department of Physiology, College of Medicine & Health Sciences, United Arab Emirates University, Al Ain P.O. Box 17666, United Arab Emirates; anemmar@uaeu.ac.ae (A.N.); ozazelzaki@uaeu.ac.ae (O.E.); elenazaaba@uaeu.ac.ae (N.E.Z.); javed.yasin@uaeu.ac.ae (J.Y.)

**Keywords:** urinary oxidative damage markers, antioxidants, obesity, diabetes, hypertension

## Abstract

Oxidative damage and inflammation are possible mechanisms linking obesity to diabetes and related complications. This study investigates the levels of oxidative damage markers in the urine of community free-living subjects with increased prevalence of obesity. Methods: Participants were assessed regarding clinical, anthropometric, and physical activity data at baseline and at 6 months. Blood and urine samples were taken for the measurements of oxidative markers in urine ((glutathione (GSH), thiobarbituric acid reactive substances (TBARS), pteridine, 8-isoprostane and 8-hydroxy-2′-deoxyguanosine (8-OH-dG)), metabolic and inflammatory markers, and related biochemical variables in the blood. Univariate and multiple regression analyses were used to assess the association between oxidative markers and other clinical prognostic indicators. Results: Overall, 168 participants with a complete 6-month follow-up with a mean (±SD) age of 41 ± 12 (119 (71%) females) were included in the study. In multiple regression analysis, log-transformed urinary pteridine levels were significantly correlated with log-transformed urinary GSH, 8-isoprostane, and TBARS after adjusting for urinary creatinine at both baseline and follow-up. Significant correlations were also found between oxidative damage markers and cardiovascular disease risk factors, including systolic blood pressure, HbA1c, plasma glucose, us-C-reactive proteins, total cholesterol, and HDL. Higher TBARS levels were found in males and diabetic subjects, with lower GSH in diabetic hypertensive and obese subjects, but the latter result did not reach statistical significance. We found nonsignificantly higher TBARS, 8-isoprostane, and pteridine levels in smokers compared to those in nonsmokers. All measured urinary oxidative damage markers levels were higher in obese subjects compared with normal-weight subjects, but results did not reach statistical significance. Conclusion: we found significant associations between urinary oxidative damage and metabolic risk factors, and higher levels of urinary oxidative damage markers in diabetic, hypertensive, smoker, and male subjects.

## 1. Introduction

The prevalence of obesity, diabetes, and other cardiovascular disease (CVD) risk factors is increasing rapidly and reaching epidemic levels in Gulf countries, including the United Arab Emirates (UAE) [1,2,3]. For example, the very recent report of ‘Diabetes around the world in 2021’ revealed that, in the Middle East and North Africa, 1 in 6 adults (73 million) are living with diabetes compared with the global 1 in 10 (537 million adults (20–79 years)). Furthermore, 1 in 3 adults living with diabetes in the Middle East are undiagnosed, and 1 in 7 live births are affected by hyperglycemia in pregnancy [4]. The UAE has one of the highest prevalences of obesity-related diabetes mellitus in the world [3].

Oxidative damage may be causatively linked to obesity-related complications, including insulin resistance and diabetes [5] In addition, oxidative damage may predict the development and progression of diabetes-related complications [5,6,7]. There is also some evidence that oxidative stress predates the appearance of diabetes complications [6]. For example, we recently reported that increased adiposity is associated with increased inflammation and decreased antioxidant status in obese subjects in the UAE [8].

Oxidative stress is a result of an imbalance between reactive oxygen species (ROS) formation, and enzymatic and nonenzymatic antioxidants. Oxidative stress biomarkers are relevant in evaluating certain diseases and the health-enhancing effects of antioxidants found mainly in dietary fruits and vegetables [9]. ROS can oxidize lipids, proteins, and nucleic acids. For example, lipid oxidation generates hydroperoxides that produce a range of end products, including 8-isoprostane, which released in the blood, and is rapidly metabolized and excreted as free acid in urine [10]. Oxidative stress to DNA is measured by the urinary excretion of 8-hydroxy-2′-deoxyguanosine (8-OHdG). The urinary excretion of the 8-OHdG biomarker for the oxidative damage of DNA is a measurement of whole-body oxidative stress that is increased in diseases such as diabetes [10,11]. Pteridine derivatives are intermediate metabolites of folic acid and its cofactors, and their levels in urine modulate oxidative stress and are associated with oxidative stress markers [12]. Furthermore, neopterin in serum and urine were suggested to be an indirect estimate of the degree of oxidative stress in patients with cancer. It is not yet known, however, if serum and urinary pteridine levels are affected in other patients with increased risk of oxidative stress, including those with obesity and diabetes [12,13]. In addition, the antioxidant glutathione (GSH) affected weight loss in patients with metabolic syndrome, including obese and diabetic patients [14].

There is, however, a lack of consensus regarding the validation, standardization, and reproducibility of methods for the measurement of ROS modifications of lipids, DNA, and proteins. Furthermore, the clinical benefits of the routine measurement of both enzymatic and nonenzymatic molecules needed to prevent ROS-induced cellular and tissue damage are still unclear. We recently conducted a number of studies with variable results on the relationship among conventional antioxidants, oxidative damage, and inflammation in obese and diabetic subjects [8,15]. A recent pilot study revealed that urinary fluorescence caused by pteridine was potently associated with oxidative stress and smoking [12]. The spectrofluorometric estimation of urinary pteridine may, therefore, be a simple and useful method for the evaluation of oxidative stress.

The aims of this study were to: (1) evaluate the utility of urinary pteridine levels as a marker of oxidative stress, and (2) examine if pteridine or other markers of oxidative stress correlate with risk factors for oxidative damage in community free-living subjects with increased prevalence of obesity. 

## 2. Methods

Free-living UAE citizens and other Arabs from neighboring countries living in the UAE were included in this study. Subjects aged 18 years and over from the city of Al Ain with a total population of 600,000 were recruited by advertisement through the local press, and from community health centers and hospital outpatient clinics. Individuals with serious physical or psychiatric illness and those who were unable to give informed written consent were excluded. Following informed written consent, eligible subjects’ blood and urine samples were taken for measurements of metabolic risk markers and urinary oxidative damage markers, respectively. Clinical assessment that included demographic and baseline characteristics, general and self-rated health, and physical activity was performed at baseline. 

### 2.1. Anthropometric and Clinical Measurements

Height and body weight were measured with subjects wearing light clothes. Body mass index (BMI) was calculated as weight in kilograms divided by the square of height in meters. Using WHO sex-adjusted cut-off points for BMI, subjects with BMI = 18.5–24.9 were classified as normal, BMI = 25.0–29.9 as overweight, and those with BMI ≥ 30 as obese. Systolic blood pressure (SBP) and diastolic blood pressure (DBP) were measured using standard methods in a sitting position after at least 5 min of rest, and the average of two readings was recorded. 

### 2.2. Biochemical Analysis

Subjects provided a fasting morning blood sample, and a spot urine sample was then also obtained for measurements of markers of oxidative damage.

Lipid peroxidation was spectrophotometrically determined using the TBARS method and the malondialdehyde (MDA) standard curve [16]. The levels of GSH were quantified using Elman’s reagent and the GSH standard curve according to the protocol from Sigma-Aldrich Fine Chemicals (St Louis, MO, USA) [16].

Concentrations of 8-isoprostane and 8-OH-dG in urine were assessed according to the manufacturer’s instructions provided in commercially available assay kits obtained from Cayman Chemicals (Ann Arbor, MI, USA). The urine concentration of the oxidized form of pteridine, neopterin, was estimated as per a previously described method [12]. Urine was diluted at 1:50 with 10 mM HEPES prior to the experiment, and the standard curve was constructed using neopterin at various concentrations (0, 6.25, 12.5, 25, 50, and 100 µM). Reading was conducted spectrofluorimetrically with an excitation of 360 nm and an emission wavelength of 450 nm [12]. Ultrasensitive C-reactive proteins (us-CRP) levels in the plasma were measured using commercial enzyme immunoassays. Other biochemical variables, including LDL and HDL cholesterol, triglycerides, blood sugar, and creatinine concentrations in serum, were measured using conventional methods. In order to evaluate renal function, plasma and urinary creatinine were also measured.

### 2.3. Statistical Analysis

Statistical analysis was performed with SPSS software version 25.0 (SPSS Inc., Chicago, IL, USA). Log-transformed data were analyzed, and geometric means (SD) are presented. A two-sample t-test and Pearson’s correlations were used. One-way ANOVA or the nonparametric Kruskal–Wallis H test was used to test for within- and between-group differences, and a *p*-value < 0.05 was considered to be significant. Multiple regression analyses were used to assess the association between oxidative markers and other clinical prognostic indicators including age, gender, smoking physical activity, body mass index, hypertension, and type 2 diabetes.

## 3. Results

### 3.1. Characteristics of Study Population

Overall, 168 participants with a complete 6-month follow-up were included in the final analysis. Table 1 shows clinical and metabolic characteristics of the study population. Using WHO cut-off points for BMI, 40 (24%) subjects had normal BMI, 54 (32%) were overweight, and 74 (44%) were obese at baseline. 

### 3.2. Oxidative Damage Marker Levels at Baseline and Follow-Up

Figure 1 shows baseline and follow-up urinary oxidative damage markers of GSH, TBARS, pteridine, 8-isoprostane, and 8-OH-dG. Statistically significant difference between baseline and follow-up results was only observed for 8-OH-dG.

### 3.3. Correlation between Baseline Urinary Pteridine and Baseline Urinary GSH, 8-Isoprostane, 8-OH-dG and TBARS

As shown in Figure 2 and Figure 3, in univariate analysis, log-transformed baseline urinary pteridine levels were significantly correlated with log-transformed urinary TBARS, GSH, and 8-isoprostane both at baseline and follow-up (*p* < 0.05). 

Table 2 shows that, in multiple regression analysis, log-transformed urinary pteridine levels were significantly correlated with log-transformed urinary GSH, 8-isoprostane, and TBARS after adjusting for urinary creatinine both at baseline and follow-up (*p* < 0.05).

### 3.4. Correlation between Cardiovascular Disease Risk Factors and Oxidative Damage Markers

As shown in Table 3, both GSH and TBARS were significantly correlated with SBP (Pearson’s correlation coefficients: 0.296 (*p* = 0.015) for GSH and 0.337 (*p* < 0.001) for TBARS). TBARS and 8-OH-dG showed significant correlations with both HbA1c and glucose. Pteridine showed significant correlation with us-CRP (r = 0.178 (*p* = 0.021)).

Table 4 shows that there were no significant correlations between changes in GSH, TBARS, pteridine, 8-isoprostane, and 8-OH-dG compared with changes in body weight, SBP, and HbA1c. The only significant correlation was between changes in TBARS and body weight after adjusting for sex (r = 0.172 (*p* = 0.28)).

### 3.5. Differences in Oxidant Damage Markers between Male and Female, Diabetic, and Hypertensive Subjects, and Healthy Subjects

Table 5 shows significantly higher TBARS and lower 8-OH-dG in male subjects compared to female subjects (*p* < 0.05). Diabetic subjects had significantly higher TBARS compared with nondiabetic subjects (*p* < 0.05). The levels of 8-OH-dG were significantly lower in both diabetic and hypertensive subjects. GSH levels were lower in diabetic and hypertensive subjects, but the result did not reach statistical significance.

### 3.6. Relationship between Smoking, BMI, and Physical Activity, and Oxidant Damage Markers

TBARS, 8-isoprostane, and pteridine levels were nonsignificantly higher in smokers compared to in nonsmokers (Table 5). All measured urinary oxidative damage markers levels were also higher in obese subjects compared with normal-weight subjects; however, results did not reach statistical significance (Table 5).

### 3.7. Multivariate Analysis of Urinary Markers Most Correlated with Clinical and Metabolic Risk Factors, and Other Prognostic Clinical Indicators

Multivariate analysis revealed significant and independent association between TBARS and gender (regression coefficient (95%): −66 (−128, −4), *p* = 0.037), diabetes (165 (97, 233), *p* < 0.001) and hypertension (85 (−160, −9), *p* = 0.028) after adjusting for age, physical activity, smoking. BMI, urinary creatinine, and history of diabetes and hypertension (Table 6). 

## 4. Discussion

We estimated on two occasions a number of oxidative markers in urine of community free-living subjects, with three-quarters of them being overweight or obese. We found significant correlations between urinary pteridine, an intermediate metabolite of folic acid, and a marker of oxidative stress and other markers of oxidative stress, including GSH, 8-isoprostane, and TBARS both at baseline and follow-up. Our results also suggest an association between some urinary oxidative markers and metabolic risk factors, including blood pressure, CRP, HDL, blood glucose, and HbA1c. However, we found no significant correlations between changes in GSH, TBARS, pteridine, 8-Isoprostane and 8-OH-dG compared with changes in body weight, SBP, and HbA1c except for changes in TBARS and body weight after adjusting for sex. In contrast, we found increased TBARS levels in current smokers compared with nonsmoker and diabetic subjects compared with subjects without diabetes. The association with TBARS and male gender, diabetes, and hypertension is independent of other known clinical prognostic indicators, including age, smoking, physical activity, and BMI.

A recent study used a similar methodology and reported for the first time that the urinary fluorescence level caused by pteridines is potently associated with known oxidative stress markers such as 8-isoprostane and the DNA/RNA oxidation product 8-OH-dG [12]. Furthermore, they reported significant association with age and smoking [12]. Our results add to previous evidence that oxidative damage may be part of the mechanisms that relate obesity to increased risk of diabetes. We reported that increased adiposity is associated with decreased antioxidant status in obese subjects in the UAE [8]. We also reported the results of a small study in which we found that antioxidant supplementations of obese diabetic subjects reduced tissue inflammation and improved antioxidant status [17]. The increased dietary intake of green vegetables also reduces the risk of type 2 diabetes [18]. The results of these two studies suggest that higher fruit and vegetable intake may mitigate oxidative damage and inflammation in subjects with visceral obesity.

With the growing epidemic of obesity and related complications, including diabetes and cardiovascular diseases, in our society and similar nations, these findings may have important health implications if proven. This is because antioxidants found in fruit and vegetables, and related dietary ingredients promote health by combating oxidative damage resulting from increased free radicals, which are linked to the pathogenesis of many chronic diseases [19]. Fruit and vegetables are rich in water, fiber, vitamins including antioxidants, and minerals. They help in weight loss through their low energy content and high dietary fiber content [20]. In particular, soluble fiber in fruits and vegetables may help in slowing down digestion, and it stimulates the release of gut hormones that promote satiety such as cholecystokinin and glucagon-like-peptide 1 [21]. The accompanied reduced absorption would consequently reduce the postprandial blood glucose response, which could improve insulin sensitivity and increase fat oxidation [20]. The Global Burden of Disease study reported that diet contributes more to diseases such as obesity, diabetes, and CVD than physical inactivity, smoking, and alcohol combined do [22]. In addition, adherence to a diet with relatively large amounts of fruit and vegetables reduces chronic diseases, and significantly decreases CVD and total mortality [23,24]. 

Our study results showed nonsignificantly higher oxidative damage markers in obese subjects compared to those in normal-weight subjects. Although BMI is an easily accessible measure of excessive body weight in the general population, it cannot differentiate between fat and lean body mass. Individuals with normal weight can still demonstrate harmful adiposity traits. Our study only considered BMI as a measure of adiposity. Another possible explanation is the relatively younger age of the study participants. For example, 75% of the UAE population are obese or overweight, with the 20% prevalence of diabetes rising to 40% among people above 40 years of age [3]. Although pteridine,8-isoprostane and 8-OH-dG were reported to be higher in smokers than those in nonsmokers [25,26], we only found nonsignificantly higher TBARS, pteridine, and 8-OH-dG levels in current smokers compared to those in nonsmokers. In addition to a poorer diet, people who smoke are also exposed to free radicals produced by cigarette smoking, leading to oxidative damage [27,28]. Several studies showed that antioxidant micronutrients such as vitamins A, C, and E, and folic acid, which are lower in current smokers, have protective effects against cigarette-smoke-induced toxicity [28]. However, our results could be explained by the under-reporting of smoking leading to the misclassification of current smokers as ex-smokers [29].

Although there is lack of consensus regarding the validation, standardization, and reproducibility of methods for measurement of ROS and their clinical benefits, urinary markers such pteridine and 8-OH-dG were recently reported to be whole-body oxidative stress and GSH that are useful in the weight management of obese and diabetic patients [10,14]. Furthermore, the urinary markers used in our study represent integrated indices of redox balance over a longer period of time compared to blood levels, which may render them more sensitive to predicting chronic conditions while decreasing intraindividual measurement variability [30]. They can also be useful in large clinical studies because of the noninvasive collection and storage [30]. We followed standard methods in our measurements of urinary oxidative damage markers, and adjusted for urinary creatinine in the analysis. Our study included a large sample of obese and nonobese subjects assessed on two occasions. 

## 5. Conclusions

We found significant correlations between DNA, and lipid and folic acid urinary markers of oxidative stress, both at baseline and follow-up. Our results also suggest an association between some urinary oxidative markers and metabolic risk factors. The use of these simple and reproducible urinary markers, if confirmed in future studies, may render them more sensitive to predicting responses to treatment in chronic conditions such as obesity and diabetes. This could have enormous public health implications for a reduction in obesity and its consequences in our community and globally.

## Figures and Tables

**Figure 1 antioxidants-11-00844-f001:**
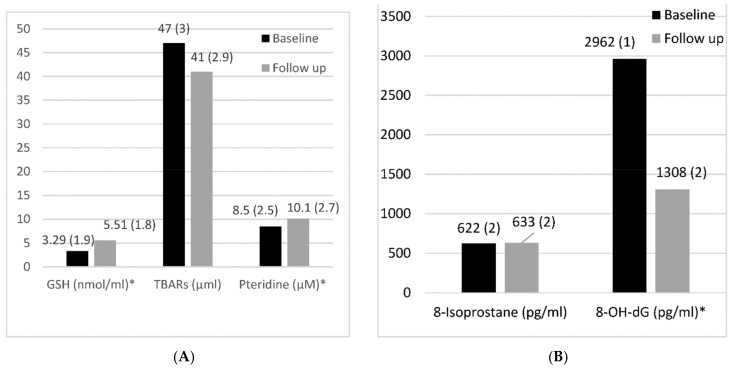
Baseline and follow-up of (**A**) urinary GSH, TBARS, and pteridine, and (**B**) 8-isoprostane and 8-OH-dG. Values represent geometric mean ± SD; * *p* < 0.05 for difference between baseline and follow-up.

**Figure 2 antioxidants-11-00844-f002:**
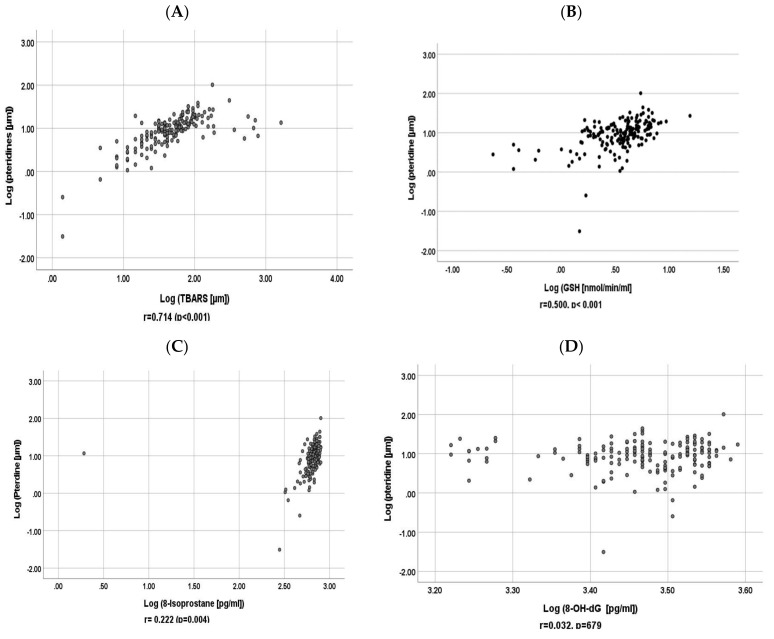
Scatter plots for correlations of log-transformed baseline urinary pteridine with log transformed baseline (**A**) urinary TBARS, (**B**) GSH, (**C**) 8-isoprostane, and (**D**) 8-OH-dG.

**Figure 3 antioxidants-11-00844-f003:**
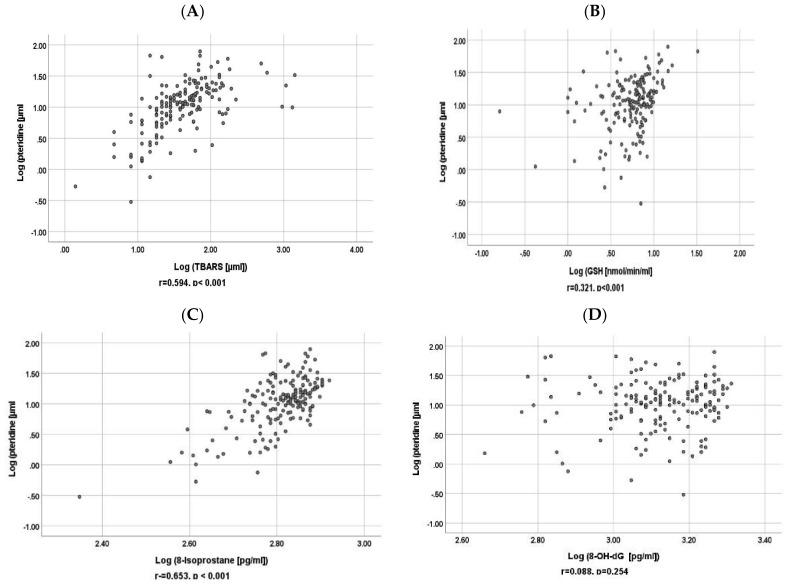
Scatter plots for correlations of log-transformed follow up urinary pteridine with log transformed follow up (**A**) urinary TBARS, (**B**) GSH, (**C**) 8-isoprostane, and (**D**) 8-OH-dG.

**Table 1 antioxidants-11-00844-t001:** Clinical and metabolic characteristics of the study population (mean (SD) unless otherwise stated).

Variables		Baseline (*n* = 168)	Follow Up (*n* = 168)
Age (years)		41 (12)	
Sex, female n (%)		119 (71)	
Smoking n (%)			
	Current	19 (11)	
	Ex-smoker	8 (5.0)	
	Never smoked	141 (84)	
Body mass index (BMI) n (%) *			
	Normal weight (BMI 18.5–25)	40 (24)	38 (23)
	Overweight (BMI 25.1–29.9)	54 (32)	53 (32)
	Obese (BMI ≥ 30)	74 (44)	73 (45)
Physical activity n (%)			
	Not active	31 (18)	20 (12)
	Moderately active	104 (62)	115 (69)
	Very active	33 (20)	32 (19)
Diabetes n (%)		33 (20)	
Hypertension n (%)		27 (16)	
Systolic blood pressure (SBP; mm Hg)		123 (13)	121 (10)
Diastolic blood pressure (DBP; mm Hg)		76 (8.0)	78 (8)
Hs CRP (mg/L)		3.5 (3.0)	3.8 (4)
HBA1c (%)		5.8 (0.9)	5.7 (1)
Total cholesterol (mmol/L)		4.9 (0.9)	
Urea (mmol/L)		4.1 (1.5)	

* *p* < 0.05 for the difference in BMI between baseline and follow-up.

**Table 2 antioxidants-11-00844-t002:** Correlation of (A) baseline and (B) follow-up urinary pteridine with urinary GSH, 8-isoprostane, 8-OH-dG, and TBARS in multiple regression analysis.

**2A**							
**Model**		**Unstandardized** **Coefficients**	**Standardized** **Coefficients**	***p*-Value**	**95.0% Confidence Interval for B**		
		**B**	**Std. Error**		**Beta**	**Lower Bound**	**Upper Bound**
1	(Constant)	−5.715	1.401		<0.001	−8.497	−2.933
	GSH	0.223	0.095	0.201	0.021	0.035	0.412
	8-Isoprostane	1.789	0.495	0.316	<0.001	0.807	2.772
	8-OH-dG	0.259	0.294	0.061	0.381	−0.325	0.842
	TBARS	0.350	0.076	0.396	<0.001	0.199	0.502
	Urine creatinine	0.001	0.006	0.012	0.881	−0.012	0.014
**2B**							
**Model**		**Unstandardized** **Coefficients**	**Standardized Coefficients**	***p*-Value**	**95.0% Confidence Interval for B**		
		**B**	**Std. Error**		**Beta**	**Lower Bound**	**Upper Bound**
	(Constant)	−5.861	1.199		<0.001	−8.241	−3.481
	GSH	0.239	0.105	0.167	0.025	0.031	0.447
	8-Isoprostane	2.471	0.416	0.482	<0.001	1.645	3.296
	8-OH-dG	−0.262	0.258	−0.067	0.312	−0.774	0.250
	TBARS	0.357	0.074	0.381	<0.001	0.211	0.503
	Urine creatinine	0.001	0.007	0.006	0.939	−0.014	0.015

**Table 3 antioxidants-11-00844-t003:** Correlation between metabolic risk factors and oxidative damage markers.

	GSH (nmol/mL)	TBARS (µmL)	8-Isoprostane (pg/mL)	Pteridine (µmL)	8-OH-dG (pg/mL)
SBP (mm Hg)	r = −0.296 (*p* = 0.015)	r = 0.337 (*p* < 0.001)	r = −0.550 (*p* = 0.513)	r = −0.047 (*p* = 0.596)	r = −0.104 (*p* = 0.238)
HbA1c (%)	r = −0.144 (*p* = 0.062)	r = 0.550 (*p* < 0.001)	r = 0.100 (*p* = 0.198)	r = −0.030 (*p* = 0.697)	r = −0.176 (*p* = 0.022)
Glucose (mmol/L)	r = −0.062 (*p* = 0.427)	r = 0.603 (*p* < 0.001)	r = 0.059 (*p* = 0.447)	r = 0.073 (*p* = 0.347)	r = −0.254 (*p* = 0.001)
C-reactive proteins (mg/L)	r = 0.139 (*p* = 0.072)	r = 0.094 (*p* = 0.224)	r = 0.038 (*p* = 0.662)	r = 0.178 (*p* = 0.021)	r = 0.147 (*p* = 0.056)
Total Cholesterol (mmol/L)	r = 0.004 (*p* = 0.963)	r = −0.083 (*p* = 0.286)	r = −0.032 (*p* = 0.676)	r = 0.029 (*p* = 0.705)	r = −0.110 (*p* = 0.153)
High densitylipoprotein (mmol/L)	r = −0.047 (*p* = 0.545)	r = −0.152 (*p* = 0.049)	r = 0.007 (*p* = 0.924)	r- = −0.101 (*p* = 0.192)	r = 0.077 (*p* = 0.317)

r = correlation coefficient.

**Table 4 antioxidants-11-00844-t004:** **Partial** correlation between changes in metabolic risk factors and oxidative damage markers between baseline and follow-up after adjusting for sex.

	GSH (nmol/mL)	TBARS (µmL)	8-Isdoprostane (pg/mL)	Pteridine (µmL)	8-OH-dG (pg/mL)
Body weight (kg)	r = −0.015 (*p* = 0.844)	r = 0.172 (*p* = 0.028)	r = 0.016 (*p* = 0.834)	r = 0.019 (*p* = 0.209)	r = −0.030 (*p* = 0.699)
SBP (mm Hg)	r = −0.103 (*p* = 0.398)	r = 0.209 (*p* = 0.089)	r = −0.005 (*p* = 0.966)	r = −0.003 (*p* = 0.979)	r = −0.054 (*p* = 0.652)
HBA1c (%)	r = −0.019 (*p* = 0.811)	r = 0.036 (*p* = 0.655)	r = 0.166 (*p* = 0.139)	r = 0.166 (*p* = 0.139)	r = −0.043 (*p* = 0.586)

**Table 5 antioxidants-11-00844-t005:** Baseline urinary oxidative damage markers levels (geometric mean (SD)) according to baseline clinical characteristics.

		GSH (nmol/mL)	TBARS (µmL)	8-Isoprostane (pg/mL)	Pteridine (µmL)	8-OH-dG (pg/mL)
Gender	Male	3.4 (1.6)	66 (3.3)	670 (1.2)	8.5 (2.7)	2754 (1.2)
	Female	3.2 (2.0)	40 (2.6) *	603 (1.8)	8.5 (2.5)	3019 (1.4) *
Smoking	Current	3.7 (1.5)	70 (3.4)	657 (1.2)	10.6 (2.1)	2842 (1.2)
	Ex-smoker	3.24 (1.8)	38 (3.2)	650 (1.1)	6.9 (2.0)	2977 (1.2)
	Never smoked	3.23 (1.9)	44 (2.8)	615 (1.7)	8.3 (2.7)	2975 (1.4)
Body mass index	Normal weight	3.4 (1.8)	42 (3.0)	566 (3.0)	7.7 (3.0)	3013 (2.0)
	Overweight	3.8 (1.8)	54 (3.0)	630 (2.0)	9.3 (2.0)	2791 (2.0)
	Obese	3.2 (1.9)	45 (2.8)	649 (2.0)	8.5 (2.6)	3061 (1.5)
Diabetes +	No	3.4 (1.9)	41.5 (1.1)	614 (1.7)	8.6 (2.7)	3019 (1.4)
	Yes	2.9 (2.0)	74 (3.7) *	655 (1.1)	8.1 (2.0)	2708 (1.2) *
Hypertension +	No	3.4 (1.9)	46 (3.0)	615 (2.0)	8.7 (2.7)	3083 (2)
	Yes	2.8 (1.6)	47 (2.3)	650 (1)	7.6 (2)	2616 (1.3) *

+ previously diagnosed diabetes mellitus or hypertension. * *p* < 0.05.

**Table 6 antioxidants-11-00844-t006:** Multiple regression results of age, gender, physical activity, BMI, history of smoking, diabetes, and high blood pressure on TBARS.

Mod		Unstandardized Coefficients		Standardized Coefficients	*p* Value	95.0% Confidence Interval for B	
		B	Std. Error	Beta		Lower Bound	Upper Bound
1	Age (years)	−0.588	1.327	−0.041	0.659	−3.209	2.034
	Gender (male/female)	−66.231	31.474	−0.180	0.037	−128.418	−4.044
	Physical activity (very active/moderately active/inactive)	−5.577	8.297	−0.049	0.502	−21.970	10.816
	Smoking (current/ex/never)	−33.208	20.294	−0.131	0.104	−73.304	6.888
	Diabetes (yes/no)	165.406	34.343	0.392	<0.001	97.550	233.261
	Hypertension (yes/no)	−85.086	38.339	−0.189	0.028	−160.836	−9.336
	Body mass index	0.937	2.532	0.029	0.712	−4.066	5.940

## Data Availability

Data is contained within the article.

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
