# Peer review of "Urinary Oxidative Damage Markers and Their Association with Obesity-Related Metabolic Risk Factors"

_antioxidants, 2022, doi:10.3390/antiox11050844_

Round 1

Reviewer 1 Report

The work has many typos, and shortcomings, which makes it very difficult to read. In addition, there is no numbering of lines. There is still no evident gap in knowledge in the introduction. The result part is described in a laconic way and is hard to understand. It takes a lot of time to understand the results, which readers may not want to spend. The discussion is poorly written. The conclusions do not correspond to the objectives of the work.

  1. 8-Hydroxy-2'-deoxyguanosine is not a DNA marker
  2. Table 1. What does the asterisk mean? For glucose, there is a semicolon in SD instead of a dot.
  3. Figure 1a: should be 'baseline and follow-up'. Please comment on the results. Please add data on the clinical and metabolic characteristics of the study population during the follow-up period (and comment on whether statistical differences were noted).
  4. Sometimes the value of p is denoted as p, sometimes as P.
  5. Figure 2,3. The r and p values are faintly visible. The axis signatures are also not visible. Please correct. Please comment on the results in the text. Did pteridine correlate with any other variables both at baseline and after follow-up?
  6. The p-value of 000 should be changed to p<0.001
  7. The results are poorly described. Reading the text alone, it is difficult to see what results were obtained g. “Table 3 shows significant correlations between GSH and TBARS with SBP, TBARS and 8-OH-dG with both HbA1c and glucose, pteridine with us-CRP and TBARS with HDL. Table 4 shows no significant correlations between changes in GSH, TBARS, pteridine, 8-Isoprostane and 8-OH-dG compared with changes in body weight, SBP and HbA1c except for changes in TBARS and body weight after adjusting for sex.” What are the differences between the tables? What is the conclusion?
  8. Table 3. Did you use Spearman or Pearson correlations?
  9. I do not understand why only body weight, SBP and HBA1c were selected for partial correlation?
  10. “Multivariate analysis revealed a significant and independent association between TBARS and gender, diabetes and hypertension after adjusting for age, gender, physical activity, smoking. BMI, urinary creatinine and history of diabetes and hypertension (p<0.05).” Association of gender-adjusted for gender?
  11. “Our results also suggest an association between some urinary oxidative markers and metabolic risk factors including blood pressures, CRP, HDL, blood glucose and HbA1c.” At this point, the results of the partial correlation should be presented
  12. “Furthermore, we also found increased TBARS levels in current smokers compared with non-smoker and diabetic subjects compared with subjects without diabetes.” The results for smoking were not significant as stated in one of the presented tables.
  13. Write why multiple regression was performed for TBARS or simply remove it.
  14. “Our results add to previous evidence that oxidative damage may be part of the mechanisms that relate obesity to increased risk of diabetes” Why? None of the markers correlated with BMI?
  15. “The results of these two studies suggest higher fruits and vegetable intake may mitigate oxidative damage and inflammation in subjects with visceral obesity.” If there is so much evidence, why has diet not been analysed here?
  16. „Although pteridine,8-isoprostane and 8-OH-dG were reported to be higher in smokers than non-smokers [25,26], we only found non-significantly higher TBARS, pteridine and 8-OH-dG levels in current smokers compared to non-smokers.” Why? No significant differences were noted in the table
  17. “Although there is a lack of consensus regarding the validation, standardization and reproducibility of methods for measurement of ROS and their clinical benefits, recently urinary markers such pteridine and 8-OH-dG have been reported to be whole-body oxidative stress and GSH useful in weight management of obese and diabetic patients” Please add the reference
  18. “Furthermore, the urinary markers used in our study represent integrated indices of redox balance over a longer period of time compared to blood levels which may make them more sensitive to predicting chronic conditions while also decreasing intra-individual variability of measurement” Please add the reference. This is not a conclusion based on your work. Furthermore, there were significant differences in urinary OS marker concentrations at baseline and follow-up.

Reviewer 2 Report

Dear authors,

I went through your work focusing mainly on introduction and discussion. I agree with the modifications and I am satisfied with your corrections in the text. What I noticed at speed are still the lack of uniformity of the figures, graphs and tables, example: page 5 figure 2B) the auxiliary x-axes are missing; figure 3 there is a bad font; tables misalignment and text centering; etc. As it is only a "formal" beauty of the text, I will not forget it anymore and I will ask authors, be careful, this is your business and scientific point in the form of precision.

Round 2

Reviewer 1 Report

The results are described very poorly. It is still difficult to understand what is being done and why.

Author Response

Reply to Reviewer 1

The results are described very poorly. It is still difficult to understand what is being done and why.

Results section rewritten with new subheadings provided to address the above concerns (Page 3-10)

I addition new text and a reference (13), added to beef up the introduction (Page 2, lines 16-19)

This manuscript is a resubmission of an earlier submission. The following is a list of the peer review reports and author responses from that submission.

Round 1

Reviewer 1 Report

The article may present interesting results, but there are a lot of them, the next steps of the procedure and their reasons are not explained, and the reader gets lost and does not understand why it is all done and what the conclusions are. Thus, the article should be rewritten.

  1. The knowledge gap is not clearly presented. There are no data on whether body fluids were previously analyzed in " community free-living subjects with increased prevalence of overweight and obesity"? What were the results? Why is it important to conduct such an analysis?
  2. „The aims of this study were therefore to investigate levels of oxidative markers including pteridine in the urine of community free-living subjects with increased prevalence of overweight and obesity and whether urinary oxidative markers correlate with metabolic risk factors.” In my opinion, that is not the goal. I see no reason to do it. Perhaps it should be to: 1) evaluate the utility of urinary pteridine levels as a marker of oxidative stress 2) see if pteridine or other markers of oxidative stress correlate with risk factors for oxidative damage. But why do this? What research is published on this topic?
  3. I agree that it is better to sample twice to eliminate within-person variability. However, there is a statistical difference for GSH, pteridine and 8-OH-dG. In the case of the latter, it is twice as small. How can this be explained?
  4. Have you verified that urinary glucose levels in diabetic patients do not affect the TBARS score?
  5. There are many typos in the manuscript
  6. Were there more diabetics in the male group? Maybe this is the reason for the gender differences
  7. Why TBARS was used in multiple regression when you were interested in pteridine. The scheme of things is not clear to me.
  8. „diabetic subjects compared with subjects without diabetes” - means the oxidative stress parameters or only TBARS?
  9. “The association with TBARS and male gender, diabetes and hypertension is independent of other known clinical prognostic indicators” Prognosis of what?
  10. “Furthermore, the urinary markers used in our study represent integrated indices of redox balance over a longer period of time compared to blood levels which may make them more sensitive to predicting chronic conditions while also decreasing intra-individual variability of measurement.” It is your conclusion?
  11. “The use of these simple and reproducible urinary markers if confirmed in future studies may make them more sensitive to predicting response to treatment in chronic conditions such as obesity and diabetes.” There is no evidence. It is not justified enough.

Reviewer 2 Report

Dear authors,

The added parts meet my requirements.

Nevertheless, many stylistic errors need to be corrected.

Abstract - you have lost a completely introductory text

Abstract line 8 - extra space

Key words - consider adding word parts - urinary and markers

Page 2, paragraph 1, line 5 - misaligned paragraph

Page 2, paragraph 2, line 8-13 - bad font and line spacing

Page 2, paragraph 4, line 7 - space extra

Page 2 paragraph 5 line 3-4 - the text is written as follows BMI space = space number space dash long space number

Page 2, paragraph 5, line 4 BMI ≥spaces 30

Page 3, paragraph 3, line 5 - after: space

Page 3 paragraph 4 line 5 - 5 word value delete

Adjust all tables and figures according to this basis and according to the word attachment

Page 3 Table 1 - edit description (mean SD unless .....) - it will fit into one line

Page 3 Table 1 - Variables. There is no dot at the end

Page 4 Table 1 - diabetes 33 (20%) - percentages are not written

Page 4 Table 1 - Hs CRP 3.5 mg - not written

Page 4 figure 1 - I would recommend bar graphs in white and black

Page 4 figure 1 - according to the appendix in the word

Page 5, paragraph 1, line 6 - gap up (small p <0.05)

Page 5-8 - edit by word

Page 6 description of the figure, two dots at the end

Page 9-11 of the table can be modified according to the word pattern

Page 9 line 1 baseline (A) and follow up (B)

In the tables pay attention to the different fonts (TBARS and uml), the statistical significance of p is written with a lowercase letter! Please add 0 at the beginning of the numbers, as they must be used since you use numbers such as 2.264, so SD must be 0.002. Align columns in tables, rows in tables!

Page 10 - below the Ccorrelation coefficient table

Page 10 Table 5 - * p <0.05 put in the description of the table similar to what I sketched in word.

Page 10 Table 5 - why is a different font used in the table than in the others? (unite in all!)

Page 13, paragraph 1, line 3 - there is no dot at the end

Page 13 paragraph 2 line 2 - (NP-17-11) is in another Font.

Page 13 - References to unify the writing of citations, as once at the end of the dot (8,9,10) another is not (5,6,7).
